# Changes in the Phenotype of Intramural Inhibitory Neurons of the Porcine Descending Colon Resulting from Glyphosate Administration

**DOI:** 10.3390/ijms242316998

**Published:** 2023-11-30

**Authors:** Michał Bulc, Jarosław Całka, Katarzyna Palus

**Affiliations:** Department of Clinical Physiology, Faculty of Veterinary Medicine, University of Warmia and Mazury in Olsztyn, Oczapowski Str. 13, 10-718 Olsztyn, Poland; calkaj@uwm.edu.pl (J.C.); katarzyna.palus@uwm.edu.pl (K.P.)

**Keywords:** glyphosate, enteric neurons, pig, descending colon, immunofluorescence

## Abstract

Environmental contamination and the resulting food contamination represent a serious problem and pose a major threat to animal and human health. The gastrointestinal tract is directly exposed to a variety of substances. One is glyphosate, whose presence in the soil is commonly observed. This study demonstrates the effects of low and high glyphosate doses on the populations of intramural neurons of the porcine descending colon. An analysis was performed on neurons ex-pressing the vasoactive intestinal peptide, pituitary adenylate cyclase-activating peptide, a neuronal isoform of nitrogen oxide synthase, and galanin. Even a low dose of glyphosate increased the number of neurons immunoreactive against the studied substances. However, the changes depended on both the plexus analysed and the substance tested. Meanwhile, a high glyphosate dose resulted in quantitative changes (an increase in the number) within neurons immunoreactive against all the studied neuropeptides/enzymes in the myenteric plexus and both submucosal plexuses. The response of the enteric nervous system in the form of an increase in the number of neurons immunoreactive against neuroprotective substances may suggest that glyphosate has a toxic effect on enteric neurons which attempt to increase their survivability through the released neuroprotective substances.

## 1. Introduction

Glyphosate is classified as a phosphonate herbicide. In terms of its chemical structure, it is a derivative of phosphonic acid bonded to a glycine molecule [1]. Currently, a molecule of this compound is subject to numerous modifications aimed at increasing the effectiveness of this herbicide. It is most commonly used in the form of isopropyl salt [2]. Moreover, the compound is characterised by very good solubility in water while being insoluble in organic compounds, including fats [3]. The herbicidal properties of glyphosate have been known for over 50 years, and the compound is now the most widely used herbicide worldwide. This is due to its high efficiency, non-selectivity, and low production costs. Such widespread use of this compound can lead to uncontrolled use and difficulties in determining the quantities of this substance in soil, edible plants, and animal tissues [4]. Glyphosate was designed to be toxic to plants, but its effects on animal tissues have been studied in various centres. Research has mainly focused on rats and rabbits, though humans have also experienced poisoning from the compound [5,6,7]. 

The gastrointestinal tract is most exposed to contact with numerous substances, including potentially toxic compounds found in food. It is also the main route for glyphosate to enter the animal and human body [8]. Due to its low fat solubility, glyphosate passes through all gastrointestinal tract sections, including the descending colon [9]. Due to its relatively large size and function, the gastrointestinal tract is subject to strict control by the nervous system in conjunction with the endocrine system. Neural control of gastrointestinal function is carried out by innervation specific to this organ [10]. More precisely, like most organs, individual sections of the gastrointestinal tract are innervated by the sympathetic and parasympathetic parts of the nervous system, with the parasympathetic part playing a significant role [11]. In addition, all sections of the gastrointestinal tract, from the oesophagus to the rectum, are innervated by the enteric nervous system, which controls the most important gastrointestinal functions (motor, secretory, and excretory activity) independently of the autonomic nervous system [12,13]. At the cellular level, the main element that builds up the enteric nervous system is neurons, whose arrangement within the gastrointestinal wall is strictly ordered and similar across animal species [13]. These neurons group into larger clusters referred to as plexuses that are connected by processes. The plexuses located between the longitudinal and the circular layer of the myenteric layer are referred to as the myenteric plexus, which is indivisible along the entire length of the gastrointestinal tract. In large animals like pigs, the submucosal plexus is located between the submuscular layer and the muscular layer of the mucosa. It is divided into the internal and external submucosal plexuses within the small and large intestines [14,15]. Like neurons of the central and peripheral nervous systems, the enteric nervous system synthesises and releases a whole spectrum of biologically active substances, which act mainly via auto- and paracrine mechanisms to regulate several gastrointestinal functions [10]. The inhibitory neuron class includes neurons that synthesise the vasoactive intestinal peptide (VIP). VIP is present in the submucosal and myenteric plexuses of each gastrointestinal tract section. Its connection with one of the three classes of receptors results in hyperpolarisation and relaxation of the muscular layer [16,17]. A similar inhibitory effect on the muscular layer of the gastrointestinal tract and blood vessels is exerted by the pituitary adenylate cyclase-activating polypeptide (PACAP) [18]. Another inhibitory transmitter is the gaseous neurotransmitter nitrogen oxide, whose synthesis involves one of three enzymes. Regarding neurons, the dominant form is the neuronal form of the nitrogen oxide synthase (nNOS) [19]. Galanin (GAL) also exerts an inhibitory effect, particularly in the gastric region. It should be added that in addition to typical functions associated with motor or secretory activity, these compounds are characterised by neuroprotective properties and thus prevent excessive degeneration of enteric neurons [20]. 

Most of the toxicological studies on glyphosate have been conducted on rats [5,6,21]. This study performed testing using a porcine model. In its physiology, the pig shows great similarity with human physiology. In particular, the physiology of the porcine gastrointestinal tract, due to the omnivorousness of the pig, is partly similar to the physiology of the human gastrointestinal tract [22,23]. Therefore, this study aimed to determine the effect of glyphosate administered at a low and high dose on the population of intramural neurons of the colon, immunoreactive against selected biologically active substances, mainly those of an inhibitory nature. The obtained results will determine the extent to which colonic neurons are exposed to the adverse effects of glyphosate.

## 2. Results

All pigs survived the duration of the experiment in good general condition. No gastrointestinal disorders such as diarrhea, constipation, or decreased appetite and changes in body weight gain between pigs from the control and experimental groups were observed. 

The presence of the studied substances was observed in all three plexuses of the enteric nervous system of the descending colon. The number of neurons expressing the test substance differed between the individual plexuses and varied between the studied groups (Table 1).

VIP-positive neurons in the control group in the myenteric plexus accounted for 8.56 ± 0.33% of the Hu C/D-positive neutron population (Figure 1A). In the submucosal plexuses, their number was slightly higher and amounted to 10.34 ± 0.47% in the outer submucosal plexus (Figure 1D), and to 12.77 ± 0.89% in the inner submucosal plexus (Figure 1G), respectively. A low glyphosate dose supplementation revealed no changes in the number of VIP-positive neurons in the myenteric plexus. However, in the submucosal plexuses, an increase in the number of VIP-positive neurons was observed to a value of 16.07 ± 0.72% in the outer plexus and to 17.83 ± 1.05% in the inner plexus (Figure 1E,H). On the other hand, glyphosate administered at a high dose increased the number of VIP-positive neurons in all the studied plexuses compared to the control group animals. In the myenteric plexus, the number of VIP-positive neurons increased to a value of 19.43 ± 1.88% (Figure 1C), while in the submucosal plexuses, it reached 23.14 ± 0.89% in the inner plexus, and 21.87 ± 1.33% in the outer submucosal plexus (Figure 1F,I). 

PACAP-positive neurons in the control animal group represented a sparse neuron population, with their number not exceeding 10% of the entire population of neurons labelled with Hu C/D marker in any of the plexuses. In the myenteric plexus, it amounted to 9.83 ± 0.23% (Figure 2A); in the outer submucosal plexus, to 7.54 ± 0.56%; and in the inner submucosal plexus, to 6.56 ± 0.84% (Figure 2D,G). A low glyphosate dose only resulted in an increase in the PACAP-positive neuron population within the neurons of the outer submucosal plexus, which reached a value of 12.34 ± 1.56% (Figure 2E). On the other hand, in the myenteric and the inner submucosal plexus, a low glyphosate dose caused no statistically significant changes in the population of PACAP-expressing neurons. A high glyphosate dose statistically significantly affected the PACAP expression in all the plexuses. In each of the studied plexuses, an increase in the population of studied neurons was observed. In the myenteric plexus, the percentage of these neurons was 15.88 ± 1.57% (Figure 2C); in the outer submucosal plexus, it was 22.98 ± 2.04%; and in the inner plexus, it was 13.60 ± 0.99%, respectively (Figure 2F,I). 

The population of neurons expressing nNOS was the largest group studied. In the control group, in the myenteric plexus, these neurons accounted for 31.98 ± 1.54% (Figure 3A); in the submucosal plexuses, for 33.15 ± 1.96% (Figure 3D) in the outer plexus, and in the inner plexus, for 38.29 ± 1.80%, respectively (Figure 3G). A low glyphosate dose statistically significantly increased the number of nNOS-positive neurons only in the myenteric plexus to a value of 36.65 ± 2.10% (Figure 3B), while in the submucosal plexuses, at a low glyphosate dose, no statistically significant changes were observed in this neuron population. However, a high glyphosate dose resulted in statistically significant changes within all the studied plexuses. In the myenteric plexus, the nNOS-positive neuron population increased to 44.87 ± 2.12% (Figure 3C); in the outer submucosal plexus, to 40.14 ± 1.95% (Figure 3F); and in the inner submucosal plexus, to 41.56 ± 2.77% (Figure 3I).

Similar to PACAP-positive neurons, GAL-positive neurons represented a population that did not exceed 10% of the entire population of Hu C/D-positive neurons in the control animal group. In the myenteric plexus, the percentage of GAL-positive neurons was 7.12 ± 0.80% (Figure 4A); in the outer submucosal plexus, it was 5.87 ± 1.08% (Figure 4D); and in the inner submucosal plexus, it was 9.74 ± 0.62% (Figure 4G). A low glyphosate dose increased the GAL-positive neuron population only within the inner submucosal plexus to a value of 13.99 ± 0.71% (Figure 4H) without causing statistically significant changes within the two other plexuses. However, a high glyphosate dose contributed to an increase in the number of GAL-positive neurons within all the studied plexuses. In the myenteric plexus, the population of GAL-immunoreactive neurons increased to 14.90 ± 1.06% (Figure 4C); in the submucosal plexuses, 11.98 ± 0.84% in the outer submucosal plexus, and to 19.09 ± 1.56% in the inner submucosal plexus (Figure 4F,I).

## 3. Discussion

The obtained results, for the first time, describe the quantitative changes in neurons of the enteric nervous system of the porcine descending colon as a result of oral supplementation with a low and high glyphosate dose. The literature contains study results describing the plasticity of enteric neurons, including those of the descending colon, under the influence of such substances as bisphenol A, acrylamide, mycotoxins, or non-steroidal anti-inflammatory drugs, or resulting from pathological conditions such as axotomy, diabetes mellitus, or an inflammation [24,25,26,27,28]. In each of the above cases, changes in the number of neurons of the enteric nervous system were determined by the dose of the substance applied, the studied section of the gastrointestinal tract, and the plexus type. The current study applied two glyphosate doses and investigated neuronal variability within the myenteric plexus and the outer and inner submucosal plexuses. In this study, a low dose was set at 0.05 mg per kg of body weight, with a high dose at 0.5 mg per kg of body weight. Other studies using other potentially toxic substances, e.g., acrylamide and its effect on neurons of the enteric nervous system, were also conducted using two doses of this compound [29,30]. In addition, the selection of the dose was determined based on the data of a European food safety agency. It should be stressed that due to its poor fat solubility, only 40% of glyphosate is absorbed, mainly in the duodenum and jejunum regions [8]. This results in achieving high glyphosate concentrations in the food content as well as its accumulation within the wall of individual sections of the gastrointestinal tract. This additionally promotes direct changes in the micro-environment of the intramural neurons and their adaptation to unfavourable conditions resulting from this fact. One of the possible adaptive neuronal changes is a change in their phenotype, which, in turn, is expressed by an increase or decrease in the number of neurons immunoreactive against different biologically active substances [12]. This study primarily identified changes in the number of neurons immunoreactive against neurotransmitters, i.e., those which reduce motor and secretory activity of the gastrointestinal tract or cause changes in the blood flow as a result of a relaxing effect on the muscular layer of blood vessels. In addition, studies conducted in recent years clearly show that the studied substances are characterised by neuroprotective activity towards neurons, including those of the enteric nervous system [31,32,33]. 

VIP represents neuropeptides with strong hyperpolarising properties, resulting primarily in the relaxation of muscle fibres [17]. In the above studies, even a low glyphosate dose increased the population of neurons synthesising this neuropeptide but only within the submucosal plexuses, while the population of VIP-positive neurons within the myenteric plexus remained at the statistically unchanged level as compared to the control group. However, a high glyphosate dose supplementation increased the neuron population also in the myenteric plexus. This may suggest that an increase in the glyphosate dose results in its greater accumulation in the gastrointestinal wall and an increased response of the intramural neurons, particularly VIP-positive, which results from the fact that VIP is one of the more potent neuroprotective agents within the enteric nervous system. The release of VIP into the environment can improve the survival of neurons through auto- and paracrine mechanisms. The mechanism of this action is mainly associated with the effect on the glial cells present in the enteric plexuses, which increase the secretion of pro-inflammatory cytokines under the influence of VIP [33,34,35]. 

Similar to VIP, PACAP also exhibits neuromodulatory and neuroprotective properties. In addition, it is involved in neuronal proliferation and differentiation and the axonal growth and development of glial cells. An increased PACAP expression in the CNS was noted in disorders caused by neurotoxic agents such as ethanol, kainic acid, or long-term hyperglycaemia [36,37]. In the gastrointestinal tract, cytoprotective properties of PACAP were noted in conditions involving intestinal ischaemia. In addition, increased immunoreactivity of PACAP within the ENS structures was noted in experimental axotomy and NSAID supplementation [18,26]. In the current study, a particularly high glyphosate dose also increased the population of PACAP-positive neurons in all three plexuses, while a lower glyphosate concentration administered orally only resulted in a response of neurons of the external submucosal plexus, where an increase in the PACAP neurons occurred. 

An opposite situation was observed for nNOS-positive neurons, as a low-dose supplementation increased the population of this neuron class in the myenteric plexus without affecting the nitrergic neurons in the submucosal plexuses. On the other hand, a high dose, as in the case of VIP and PACAP, increased the number of nNOS-positive neurons in all the studied plexuses. The function of nitrogen oxide within different sections of the gastrointestinal tract is mainly reduced to the inhibitory effect on motility [18,38]. Previous studies have demonstrated that this gaseous neurotransmitter causes the relaxation of the oesophagus and the stomach as well as the individual sections of the small and large intestine, including the internal anal sphincter muscle. The inhibitory effect of nitrogen oxide is not limited only to the effect on the muscular layer of the gastrointestinal layer but includes the effect on the secretion of gastrointestinal hormones and the muscular layer of blood vessels [38,39]. The results of the current study may suggest that an increase in the nNOS-positive neurons within the myenteric plexus may affect motility, as these are the neurons of the myenteric plexus that contribute, through the substances released, to the generation of contractions in the smooth and circular muscular layer of the gastrointestinal tract. An increase in the number of these neurons may intensify the inhibitory effect of nitrogen oxide, which is particularly important for the motility of the descending colon, whose contractions contribute to the defecation process.

The last substance studied was galanin. The neuronal response to a low glyphosate dose was only visible in the internal submucosal plexus, where the number of GAL-positive neurons increased. A high dose resulted in a significant increase in the population of GAL-immunoreactive neurons in all three intramural plexuses of the descending colon. Galanin, like VIP, is a substance with neuroprotective properties against neurons of both the central and the peripheral nervous systems. As for galanin-positive neurons found in the enteric nervous system, their increase was observed in the course of pathological conditions leading to neurodegeneration (colitis of both inflammatory and chemical origin, acrylamide supplementation), which confirms the neuroprotective role of galanin. The mechanism of this action, similar to VIP, is indirect through a modulatory effect on the secretion of pro-inflammatory cytokines [40,41,42]. 

The digestive tract is particularly exposed to the toxic effects of many substances; one of these is glyphosate. Studies performed in rats have shown that even a single administration of glyphosate results in its accumulation in the wall of the gastrointestinal tract [9]. This high penetration and accumulation of glyphosate is due to its properties, namely, its disruption of the production of proteins that create tight intercellular connections between intestinal epithelial cells [43]. This was observed in human intestinal epithelial cell cultures and in vivo studies using zebra fish [44,45]. In both cases, a decrease in mRNA for claudin 5, a protein responsible for creating intercellular connections, was observed. However, one of the main harmful effects of glyphosate may be its neurotoxicity. This effect has been confirmed in the case of neurons of the central nervous system [46]. One of the main mechanisms of this action is the influence on the enzymatic systems responsible for the production of free radicals, primarily superoxide dismutase and catalase [47]. Neurons of the enteric nervous system may also be exposed to glyphosate-induced oxidative stress. This may be reflected in an increase in the population of neurons immunoreactive to the tested substances. VIP, PACAP, nNOS, and GAL have neuroprotective properties. An increase in PACP and VIP expression was observed in an animal model of Alzheimer’s and Parkinson’s disease [48]. 

Therefore, in summary, it should be emphasized that glyphosate intoxication, especially in high doses, disrupts the proper functioning of neurons of the enteric nervous system. In response, neurons increase the synthesis of neuroprotective substances, which protects them from further damage. It is an early stage of the enteric nervous system response to potential damage with no clinical symptoms yet visible. Therefore, analysis of the phenotypic variability of neurons may constitute the first, preclinical stage of damage to the gastrointestinal tract by glyphosate. The exact mechanism of action of glyphosate on ENS neurons is unknown. However, the changes in the population of ENS neurons immunoreactive to the tested substances observed in the present study may result from increased synthesis of these neurotransmitters or changes in axonal transport. Although there are no data in the literature indicating that glyphosate leads to the death of neurons, further research is necessary to understand the exact mechanism of the harmful effects of glyphosate on ENS neurons.

## 4. Materials and Methods

The experimental part of the study was conducted on 15 sexually immature Danish gilts (8 weeks, approx. 20 kg b.w.). Before the start of the experiment, the animals were randomly divided into three test groups. The first group comprised control animals receiving empty gelatine capsules. In the second group, the gilts were administered glyphosate (analytical standard, purity > 99.5%, Sigma-Aldrich, Darmstadt, Germany, CAS-No.: 1071-83-6) at a low dose (which corresponds to the theoretical maximum daily intake of glyphosate allowed in Europe, i.e., 0.05 mg per kg of body weight). The animals from the third experimental group received glyphosate at a dose described as high but acceptable (0.5 mg per kg of body weight, equivalent to the acceptable daily intake (ADI) in Europe). The entire experiment lasted for 28 days. Glyphosate was administered orally in capsules once daily during the morning feeding. The animals were housed in pens adapted to this animal species during the entire experiment. The animals were fed twice daily with commercial feed adapted to this animal species. The gilts had constant access to water. The light cycle was 12/12. After 28 days, animals from all the groups were euthanized. Each animal was premedicated with azaperone, and after 15 min, the animals were given a dose of sodium pentobarbital intravenously, resulting in cardiac and respiratory arrest. Immediately after the euthanasia, 2 cm sections of the descending colon were collected from each animal. The collected tissues were assigned for further testing. The Local Research Ethics Committee approved all of the above-described procedures for animal experiments (Approval No. 62/2020). 

The collected descending colon segments were immersion-fixed in a 4% buffered paraformaldehyde solution for 1 h. The tissues were then rinsed three times in a phosphate buffer solution, replacing the buffer every 24 h. After this stage, the samples were transferred to a 30% sucrose solution and kept in the refrigerator until the tissues were completely saturated with sucrose. The prepared tissues were used to make serial freezing sections with a thickness of 12 µm, which were placed on degreased microscope slides covered with gelatine. The prepared sections were subjected to the double immunofluorescent staining procedure (as described previously [46]). On the first day, the sections on the slides were dried at room temperature for 1 h, then rinsed three times in a phosphate buffer solution, with each rinse lasting for 10 min. After rinsing, the tissues were blocked in a blocking solution for 1 h to eliminate non-specific antigen–antibody reactions. The tissues were incubated in a mixture of primary antibodies after eliminating non-specific bindings. This study used the Hu C/D marker to identify neural cells and selected biologically active substances. A full specification of the antibodies used is provided in Table 2. After 12 h incubation in a mixture of appropriate primary antibodies, the tissues were rinsed again three times in a phosphate buffer solution. Afterwards, they were incubated in a solution of appropriately prepared secondary antibodies for 2 h. A full specification of the secondary antibodies used is provided in Table 2. After completing the immunofluorescent staining procedure, the specimens were analysed using a fluorescence microscope with a digital camera and appropriate image analysis software.

A fluorescence microscope with appropriate filters, a digital camera, and software were used for analysing the staining and photographic documentation. The final result of the population of neurons immunoreactive against the studied neuropeptides was presented as a percentage of neurons in relation to neurons immunoreactive against the Hu C/D neuronal marker. In this study, the percentage of neurons immunoreactive against the substances studied was obtained by counting at least 700 Hu C/D-positive neurons for each type of intramural plexus (the MP, the OSP, the ISP) of all gilts. In addition, to avoid counting populations of the same neurons, neurons located at least 100 µm apart from each other were subjected to analysis. The obtained results were then analysed statistically using the Statistica 13 program (Stat Soft Inc., Tulsa, OK, USA) using a one-way analysis of variance (ANOVA) with Dunnett’s test and expressed as a mean ± standard error of the mean (SEM) (* *p* < 0.05, ** *p* < 0.01, *** *p* < 0.001)

## 5. Conclusions

The current study demonstrated that a 28-day glyphosate supplementation caused changes in the number of enteric neurons of the descending colon, which are immunoreactive against the studied substances. A low glyphosate dose increased the number of neurons immunoreactive against the studied substances. However, the changes depended on both the plexus analysed and the substance tested, while a high dose visibly affected the neuron population in all the intramural plexuses of the descending colon. Obviously, one needs to note various limitations arising from this type of testing, such as the supplementation time or the dose. Nevertheless, the response of the enteric nervous system in the form of an increase in the number of neurons immunoreactive against neuroprotective substances may suggest that glyphosate (at doses corresponding to environmental exposure) has a toxic effect on enteric neurons which attempt to increase their survivability through the released neuroprotective substances. These changes do not necessarily result in visible clinical gastrointestinal symptoms or run a subclinical course. Still, there is no doubt that glyphosate, especially at a high dose, is not inert to the enteric nervous system of the descending colon. The exact mechanism of the glyphosate action itself on enteric nerve cells requires further research.

## Figures and Tables

**Figure 1 ijms-24-16998-f001:**
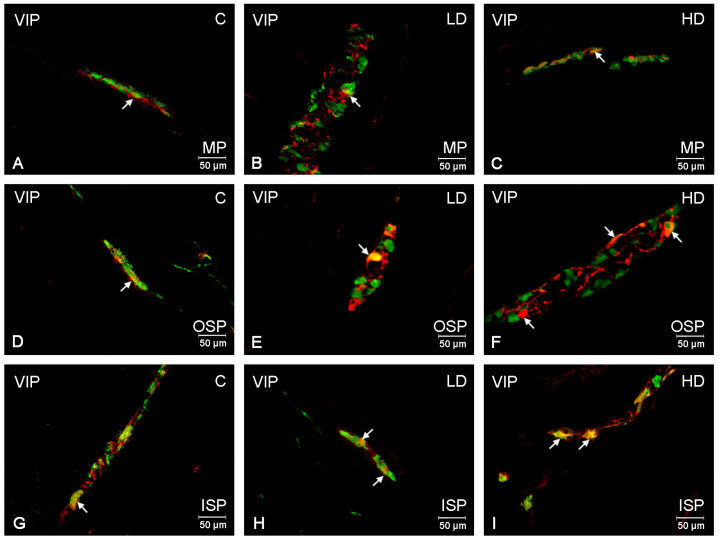
Microphotographs showing enteric neurons immunopositive to VIP in the porcine descending colon. (**A**)—neurons immunoreactive to HuC/D (panneuronal marker) and VIP in the MP of control pigs; (**B**)—neurons immunoreactive to HuC/D and VIP in the MP of low-dose group, (**C**)—neurons immunoreactive to HuC/D and VIP in the MP of high-dose group; (**D**)—neurons immunoreactive to HuC/D and VIP in the OSP of control pigs. (**E**)—neurons immunoreactive to HuC/D and VIP in the OSP of low-dose group, (**F**)—neurons immunoreactive to HuC/D and VIP in the OSP of high-dose group; (**G**)—neurons immunoreactive to HuC/D and VIP in the ISP of control pigs. (**H**)—neurons immunoreactive to HuC/D and VIP in the ISP of low-dose group, (**I**)—neurons immunoreactive to HuC/D and VIP in the ISP of high-dose group. All pictures were created by digital superimposition of two colour channels (green for HuC/D and red for VIP). MP—myenteric plexus; OSP—outer submucosal plexus; ISP—inner submucosal plexus. The arrows indicate cells bodies immunoreactive to studied substances.

**Figure 2 ijms-24-16998-f002:**
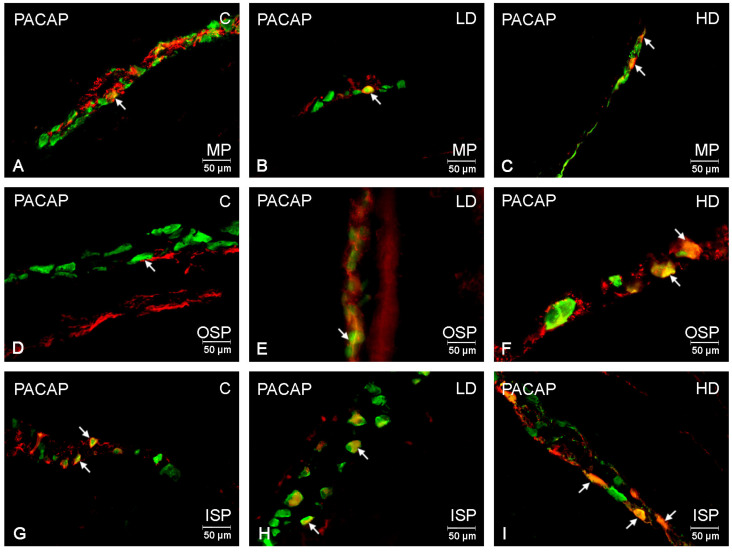
Microphotographs showing enteric neurons immunopositive to PACAP in the porcine descending colon. (**A**)—neurons immunoreactive to HuC/D (panneuronal marker) and PACAP in the MP of control pigs; (**B**)—neurons immunoreactive to HuC/D and PACAP in the MP of low-dose group, (**C**)—neurons immunoreactive to HuC/D and PACAP in the MP of high-dose group; (**D**)—neurons immunoreactive to HuC/D and PACAP in the OSP of control pigs. (**E**)—neurons immunoreactive to HuC/D and PACAP in the OSP of low-dose group, (**F**)—neurons immunoreactive to HuC/D and PACAP in the OSP of high-dose group; (**G**)—neurons immunoreactive to HuC/D and PACAP in the ISP of control pigs. (**H**)—neurons immunoreactive to HuC/D and PACAP in the ISP of low-dose group, (**I**)—neurons immunoreactive to HuC/D and PACAP in the ISP of high-dose group. All pictures were created by digital superimposition of two colour channels (green for HuC/D and red for PACAP). MP—myenteric plexus; OSP outer submucosal plexus; ISP—inner submucosal plexus. The arrows indicate cells bodies immunoreactive to studied substances.

**Figure 3 ijms-24-16998-f003:**
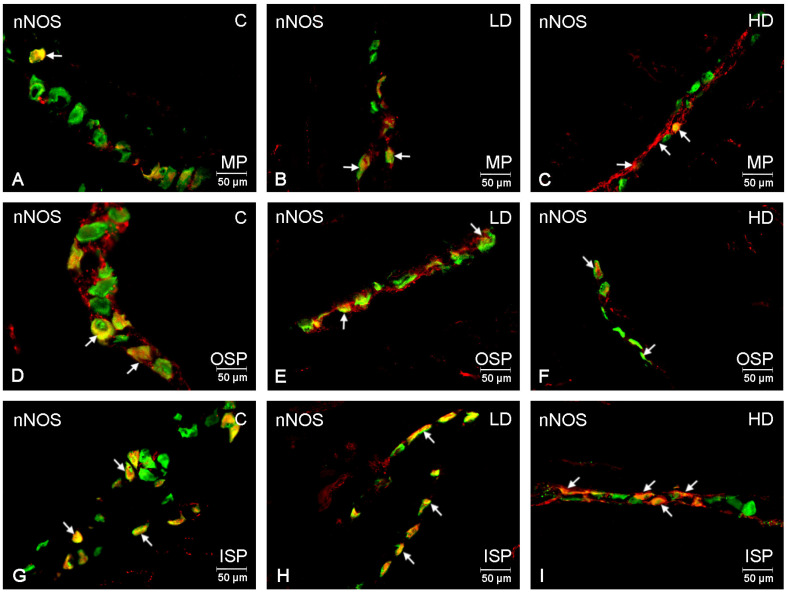
Microphotographs showing enteric neurons immunopositive to nNOS in the porcine descending colon. (**A**)—neurons immunoreactive to HuC/D (panneuronal marker) and nNOS in the MP of control pigs; (**B**)—neurons immunoreactive to HuC/D and nNOS in the MP of low-dose group, (**C**)—neurons immunoreactive to HuC/D and nNOS in the MP of high-dose group; (**D**)—neurons immunoreactive to HuC/D and nNOS in the OSP of control pigs. (**E**)—neurons immunoreactive to HuC/D and nNOS in the OSP of low-dose group, (**F**)—neurons immunoreactive to HuC/D and nNOS in the OSP of high-dose group; (**G**)—neurons immunoreactive to HuC/D and nNOS in the ISP of control pigs. (**H**)—neurons immunoreactive to HuC/D and nNOS in the ISP of low-dose group, (**I**)—neurons immunoreactive to HuC/D and nNOS in the ISP of high-dose group; All pictures were created by digital superimposition of two colour channels (green for HuC/D and red for nNOS). MP—myenteric plexus; OSP outer submucosal plexus; ISP—inner submucosal plexus. The arrows indicate cells bodies immunoreactive to studied substances.

**Figure 4 ijms-24-16998-f004:**
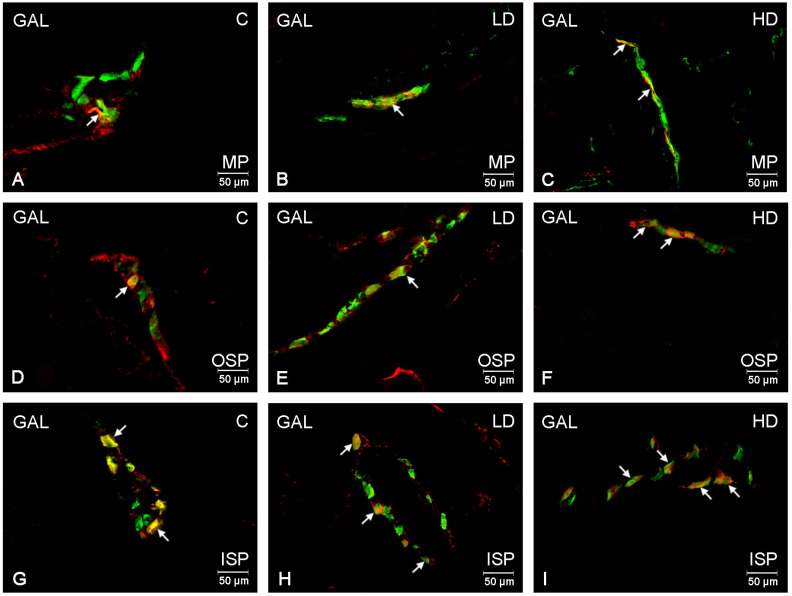
Microphotographs showing enteric neurons immunopositive to GAL in the porcine descending colon. (**A**)—neurons immunoreactive to HuC/D (panneuronal marker) and GAL in the MP of control pigs; (**B**)—neurons immunoreactive to HuC/D and GAL in the MP of low-dose group, (**C**)—neurons immunoreactive to HuC/D and GAL in the MP of high-dose group; (**D**)—neurons immunoreactive to HuC/D and GAL in the OSP of control pigs. (**E**)—neurons immunoreactive to HuC/D and GAL in the OSP of low-dose group, (**F**)—neurons immunoreactive to HuC/D and GAL in the OSP of high-dose group; (**G**)—neurons immunoreactive to HuC/D and GAL in the ISP of control pigs. (**H**)—neurons immunoreactive to HuC/D and GAL in the ISP of low-dose group, (**I**)—neurons immunoreactive to HuC/D and GAL in the ISP of high-dose group; All pictures were created by digital superimposition of two colour channels (green for HuC/D and red for GAL). MP—myenteric plexus; OSP outer submucosal plexus; ISP—inner submucosal plexus. The arrows indicate cells bodies immunoreactive to studied substances.

**Table 1 ijms-24-16998-t001:** Percentage of VIP-, PACAP-, nNOS-, and GAL-immunoreactive enteric neurons of the porcine descending colon in control and experimental groups (administrated with low and high doses of glyphosate). MP—myenteric plexus; OSP—outer submucosal plexus; ISP—inner submucosal plexus. SEM—Standard error of the mean (* *p* < 0.05, ** *p* < 0.01, *** *p* < 0.001).

Investigate Substance	Control Group			Low-Dose Group			High-Dose Group		
	MP	OSP	ISP	MP	OSP	ISP	MP	OSP	ISP
VIP	8.56	10.34	12.77	8.68	16.07 *	17.83 *	19.43 ***	21.78 ***	23.14 ***
±SEM	0.30	0.47	0.89	0.44	0.72	1.05	1.88	1.33	0.89
PACAP	9.83	7.54	6.56	9.50	12.34 **	6.49	15.88 **	22.98 ***	13.60 **
±SEM	0.23	0.56	0.84	0.88	1.56	0.93	1.57	2.04	0.99
nNOS	31.98	33.15	38.29	36.65 *	31.98	39.01	44.78 **	40.14 *	41.58 *
±SEM	1.54	1.96	1.80	2.06	1.74	2.55	2.12	1.95	2.77
GAL	7.12	5.87	9.74	6.60	5.56	13.99 *	14.90 **	11.98 ***	19.09 ***
±SEM	0.80	1.08	0.62	0.45	0.96	0.71	1.06	0.84	1.56

**Table 2 ijms-24-16998-t002:** List of primary and secondary antibodies used in the present study.

Primary Antibodies				
Antigen	Host Species	Catalogue Number	Final Dilution	Supplier
Hu C/D	mouse	A-21271	1:1000	Thermo Fisher Scientific. Waltham, MA, USA
VIP	rabbit	VA1285	1:6000	Biomol, Hamburg, Germany
PACAP	guinea pig	T-5039	1:3000	Peninsula, San Carlos, CA, USA
nNOS	rabbit	AB5380	1:2000	Sigma-Aldrich, Saint Louis, MO, USA
GAL	rabbit	AB 2233	1:1000	Millipore, Billerica, MA, USA
**Secondary Antibodies**				
Alexa Fluor 488 anti-mouse	donkey	A21202	1:1000	Thermo Fisher Scientific. Waltham, MA, USA
Alexa Fluor 488 anti-guinea pig	donkey	A11074	1:1000	Thermo Fisher Scientific. Waltham, MA, USA
Alexa Fluor 488 anti-rabbit	goat	A11010	1:1000	Thermo Fisher Scientific. Waltham, MA, USA

## Data Availability

All relevant data are contained within the manuscript.

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
