# Peer review of "Changes in the Phenotype of Intramural Inhibitory Neurons of the Porcine Descending Colon Resulting from Glyphosate Administration"

_ijms, 2023, doi:10.3390/ijms242316998_

Round 1
Reviewer 1 Report
Comments and Suggestions for Authors
The authors studied the in vivo effect of glyphosate on intramural inhibitory neurons of the porcine descending colon and demonstrated a dose-dependent increase in such neuron populations.
The manuscript is only a preliminary description of the effects without mechanistic insight, but the results are consistent and interesting, and I would be happy to recommend publication after revision of the following points.
1 It is unclear to me whether the reported effects are adverse (toxic) or not or just a biological response of the animals. Do the authors note any toxic (especially gastrointestinal) effects in dosed animals? Any changes in body weight or body-weight gain? If not, this should also be reported.
2 Please provide the following information in section 4: pig species, age, and weight at the start of the study. Please also include the source and purity of the glyphosate.
3 It is assumed 5 animals per experimental condition. Count 700 neurons per experimental condition. 700 neurons per animal (i.e., 700 x 5 = 3500 neurons per experimental condition?) Please clarify.
4 What is the error in the measurements? Is it really relevant to express the numbers in Table 1 with up to 4 significant digits?
5 According to EFSA, the acceptable daily intake is 0.3 mg/kg bw/day (file:///C:/Users/msogorb/Downloads/list1_glyphosate_en.pdf ). Thus, the highest dose in this study is almost twice the supposedly safe dose, and at real physiologically relevant exposures these effects are unlikely to pose a serious risk. This should be noted in the discussion.
6 Regarding the dose, could the dose in pigs be converted to Human Equivalent Dose (https://www.ncbi.nlm.nih.gov/pmc/articles/PMC4804402/) for a proper risk assessment?
Author Response
The authors studied the in vivo effect of glyphosate on intramural inhibitory neurons of the porcine descending colon and demonstrated a dose-dependent increase in such neuron populations.
The manuscript is only a preliminary description of the effects without mechanistic insight, but the results are consistent and interesting, and I would be happy to recommend publication after revision of the following points.
1 It is unclear to me whether the reported effects are adverse (toxic) or not or just a biological response of the animals. Do the authors note any toxic (especially gastrointestinal) effects in dosed animals? Any changes in body weight or body-weight gain? If not, this should also be reported.
The exact mechanism of action of glyphosate on ENS neurons is unknown. However, the changes in the population of ENS neurons immunoreactive to the tested substances observed in the present study may result from increased synthesis of these neurotransmitters or changes in axonal transport. Although there is no data in the literature that glyphosate leads to the death of neurons, further research is necessary to understand the exact mechanism of the harmful effects of glyphosate on ENS neurons. It is an early stage of the enteric nervous system response to potential damage with no clinical symptoms yet visible. Therefore, the analysis of phenotypic variability of neurons may constitute the first, preclinical stage of damage to the gastrointestinal tract by glyphosate.
During the experiment, no gastrointestinal disorders such as diarrhoea, constipation or decreased appetite were observed in control pigs and pigs receiving glyphosate.
All animals were weighed on the first day of the experiment and then weighed once a week in order to established an appropriate dose of glyphosate as well as compare the weight of gilts in particular groups. The obtained data were analysed statistically. There were no statistically significant differences in weights between the groups.
We added this information to the manuscript. (Lines 86-90)
2 Please provide the following information in section 4: pig species, age, and weight at the start of the study. Please also include the source and purity of the glyphosate.
According to the reviewer's suggestion all necessary information has been added in the Material and Method section.
3 It is assumed 5 animals per experimental condition. Count 700 neurons per experimental condition. 700 neurons per animal (i.e., 700 x 5 = 3500 neurons per experimental condition?) Please clarify.
To evaluate a significant population of ENS neurons in each plexus type (MP, ISP, OSP) at least 700 HuC / D cells for each plexus were counted, i.e. 2100 cells for each pig. We had 5 animal in each group so for each plexus we counted 3500 neurons. We take into account such a large number of neurons that it reflects the entire population of ENS neurons in the descending colon. We have modified the description in the materials and methods section for better understanding.
4 What is the error in the measurements? Is it really relevant to express the numbers in Table 1 with up to 4 significant digits?
In this type of research, the generally accepted rule is to present results to 2 decimal places. The data collected in Table 1 are the average of each group (average of 5 results in each group) ± standard error of the mean (SEM). In our opinion, this way of presenting the results reflects changes better than giving full percentages, because sometimes the changes are very small, e.g. 0.02%.
5 According to EFSA, the acceptable daily intake is 0.3 mg/kg bw/day (file:///C:/Users/msogorb/Downloads/list1_glyphosate_en.pdf ). Thus, the highest dose in this study is almost twice the supposedly safe dose, and at real physiologically relevant exposures these effects are unlikely to pose a serious risk. This should be noted in the discussion.
The objective of this study was to verify whether low doses of glyphosate (equivalent to the environmental exposure) bring about changes in the population of intramural neurons of the colon, immunoreactive against selected biologically active substances, mainly those of an inhibitory nature. In the article we used name “low” and “high” doses but both doses are safe for human. We used this nomenclature to emphasize that doses differ each other. The high dose is 10 times greater than the low dose.
A low dose of glyphosate: corresponds to the theoretical maximum daily intake of glyphosate allowed in Europe, i.e. 0.05 mg per kg of body weight) while a high dose of glyphosate: 0.5 mg per kg of body weight - equivalent to the acceptable daily intake (ADI) in Europe. So still both doses are “safe” for human consumption.
To better understand we added this information in the Material and Method section and conclusion section (Lines 324, 380-381).
6 Regarding the dose, could the dose in pigs be converted to Human Equivalent Dose (https://www.ncbi.nlm.nih.gov/pmc/articles/PMC4804402/) for a proper risk assessment?
The glyphosate doses used in this experiment are human doses. There are no established doses for pigs. This study was conducted on the domestic pig model, whose gastrointestinal tract is similar to humans due to its anatomy, histology and the physiological processes in it, which makes it the most suitable animal to be used in a study of the effect of food toxins on the gastrointestinal tract. However, this is still basic research, which can only be a starting point for further clinical research.

Reviewer 2 Report
Comments and Suggestions for Authors
In their paper entitled “Changes in the phenotype of intramural inhibitory neurons of the porcine descending colon resulting from glyphosate administration”, the Authors report that glyphosate, a frequent environmental contaminant, can induce porcine intramural neurons to express neuroprotective molecules. This effect is seen even at low dose of glyphosate.
Although further work is required in order to understand the molecular mechanisms underlining the observed effects of glyphosate, it is clear that this substance has an effect on the gastrointestinal system; thus, the paper is of interest and suitable for Int. J. Mol. Sci. Moreover, Methods and Results are well described, and the figures are explicative.
Major comment: Actually, the observed increase of the percentage of neurons expressing the studied substances might be due to induction of their expression in a higher number of neurons or also to a general change of neuronal number (for example, those unable to express protective molecules might die). Did the Authors counted the total amount of neurons in the different tissue sections analyzed? As the meaning of the observation might be different in the two cases, at least a comment on this point in the Discussion Section should be appropriate.
Minor comments:
Legend to Tab.1: Please explain also here (even if done in the figure legends) the meaning of OSP, ISP and MP;
Materials and Methods: pag. 9, line 309: the third group, and not the second, here.
Author Response
In their paper entitled “Changes in the phenotype of intramural inhibitory neurons of the porcine descending colon resulting from glyphosate administration”, the Authors report that glyphosate, a frequent environmental contaminant, can induce porcine intramural neurons to express neuroprotective molecules. This effect is seen even at low dose of glyphosate.
Although further work is required in order to understand the molecular mechanisms underlining the observed effects of glyphosate, it is clear that this substance has an effect on the gastrointestinal system; thus, the paper is of interest and suitable for Int. J. Mol. Sci. Moreover, Methods and Results are well described, and the figures are explicative.
Major comment: Actually, the observed increase of the percentage of neurons expressing the studied substances might be due to induction of their expression in a higher number of neurons or also to a general change of neuronal number (for example, those unable to express protective molecules might die). Did the Authors counted the total amount of neurons in the different tissue sections analyzed? As the meaning of the observation might be different in the two cases, at least a comment on this point in the Discussion Section should be appropriate.
It is not possible to give the total number of HuC/D neurons, because we do not study the isolated ganglia, but cross-sections through the intestine. HuC / D as a marker of neuronal cells is for us the total population of the studied neurons. To evaluate a significant population of ENS neurons in each plexus type (MP, ISP, OSP) at least 700 HuC / D cells for each plexus were counted, i.e. 2100 cells for each pig. This method of counting and evaluating cells is commonly used in ENS plexuses and supported by previous publications in the field. Please note that the aim of present study was the evaluation of changes in neurochemical phenotype of ENS neurons.
We agree with the reviewer that the observed changes reported reflect an effect on expression of neuropeptides that results in increased labelling of the soma or be a general change of neuronal number. However, there is no data in the literature that glyphosate, especially in low doses, leads to the death of neurons.
We added this information to the discussion:
“The exact mechanism of action of glyphosate on ENS neurons is unknown. However, the changes in the population of ENS neurons immunoreactive to the tested substances observed in the present study may result from increased synthesis of these neurotransmitters or changes in axonal transport. Although there is no data in the literature that glyphosate leads to the death of neurons, further research is necessary to understand the exact mechanism of the harmful effects of glyphosate on ENS neurons.”
Minor comments:
Legend to Tab.1: Please explain also here (even if done in the figure legends) the meaning of OSP, ISP and MP;
According to the reviewer`s suggestion the table 1 description has been corrected.
Materials and Methods: pag. 9, line 309: the third group, and not the second, here.
According to the reviewer`s suggestion correction has been done.

Reviewer 3 Report
Comments and Suggestions for Authors
The manuscript reports some very interesting results on a very controversial topic, debated between supporters and detractors, and on which much clarity has not yet been achieved, so any contribution is useful to shed light on the true situation.
The authors have a long history of research on the topic, well documented by several scientific papers, and in fact the present manuscript is well organized, with clear and precise descriptions of the objectives, results and methods used to obtain them.
Figures are very interesting and captions exhaustive.
The bibliography is relevant, although not always up to date
Overall, a very well-done job.
Author Response
The manuscript reports some very interesting results on a very controversial topic, debated between supporters and detractors, and on which much clarity has not yet been achieved, so any contribution is useful to shed light on the true situation.
The authors have a long history of research on the topic, well documented by several scientific papers, and in fact the present manuscript is well organized, with clear and precise descriptions of the objectives, results and methods used to obtain them.
Figures are very interesting and captions exhaustive.
The bibliography is relevant, although not always up to date
Overall, a very well-done job.
Thank you for your insightful review.

Round 2
Reviewer 1 Report
Comments and Suggestions for Authors
Thank you very much to the authors for reviewing the manuscript accordingly to the points that I raised in my first report. I have no further objections and suggest the publication of the manuscript in the current state.